# Scanning Algorithm Optimization for Achieving Low-Roughness Surfaces Using Ultrashort Laser Pulses: A Comparative Study

**DOI:** 10.3390/ma16072788

**Published:** 2023-03-30

**Authors:** Evaldas Kažukauskas, Simas Butkus, Vytautas Jukna, Domas Paipulas, Valdas Sirutkaitis

**Affiliations:** Laser Research Center, Faculty of Physics, Vilnius University, Saulėtekio Ave. 10, LT-10223 Vilnius, Lithuania; simas.butkus@ff.vu.lt (S.B.); vytautas.jukna@ff.vu.lt (V.J.); domas.paipulas@ff.vu.lt (D.P.); valdas.sirutkaitis@ff.vu.lt (V.S.)

**Keywords:** femtosecond pulses, laser ablation, surface roughness, scanning algorithms

## Abstract

Femtosecond laser-assisted material surface modification is a rapidly growing field with numerous applications, including tribology, micromechanics, optofluidics, and medical implant treatment. For many of these applications, precise control of surface roughness after laser treatment is crucial, as it directly affects the final properties of the work surface. However, achieving low mean surface roughness values (<100 nm) is challenging due to the fundamental principles of laser light–matter interactions. The complex physical processes that occur during laser material interactions make it difficult to achieve the desired surface roughness, and only advanced scanning methods can potentially solve this issue. In our study, we analyzed laser scanning algorithms to determine the optimal method for producing surfaces with minimal roughness. We investigated how scanning parameters such as the overlap of modifications, the amount of successive line shift, and laser-scanner synchronization impact surface roughness. Using a numerical model, we obtained results that showed good agreement with experimentally acquired data. Our detailed theoretical and experimental analysis of different scanning methods can provide valuable information for the future optimization of minimal-roughness micromachining.

## 1. Introduction

Material surface properties can be drastically varied via changes in the surface topography and roughness. There are many techniques to achieve different surface structures, including conventional abrasive polishing, chemical treatment, lithography, and others. However, the most versatile tool is laser ablation. It was shown that for metals, surface roughness determines the rate of corrosion [1], for titanium and other materials—their wettability properties [2,3,4]. In addition, surface roughness has a huge impact on fluid dynamics [5,6] and optical properties such as absorption and reflection [7,8]. Another prosperous field, where the control of surface roughness is in huge demand, is the manufacturing of various implants. It was shown that the rate of success of an implant does not depend only on the bio-compatibility properties of the material; it also depends on its surface quality [9,10,11,12]. The importance of surface roughness becomes even more pronounced when talking about vision implants, specifically intraocular lenses (IOLs). The optical properties, success of implantation, and even post-surgery complications of such lenses are greatly determined by the roughness of the lens [13]. From the given examples, it is evident that surface roughness is an important material parameter that has to be considered in many fields of applications. Hence, the ability to tailor surface roughness during the manufacturing process is of huge importance.

Due to the fast technological advancements in laser technologies and beam transport techniques, material surface treatment is more often carried out using laser light. Laser micromachining is an appealing surface treatment tool because it allows for the fast, flexible, and contactless machining of the surface. With the use of femtosecond lasers, it is possible to machine almost all material surfaces and achieve high surface quality (Ra < 1 µm [14,15]). However, achieving the optical quality of the surface (Rq < 100 nm [16]) is not an easy task even when fs pulses are considered. There is still little to no knowledge of how to control or minimize surface roughness during the laser ablation process. Usually, controlling the roughness of the surface requires conducting parametric studies to determine the necessary parameters for adjusting the surface roughness magnitude [17]. The increase or decrease in surface roughness due to a change in laser parameters (fluence, wavelength, pulse duration) is associated with different laser–matter interactions (magnitude of penetration depth, the volume of the heat-affected zone, etc.). Since some of the laser parameters, e.g., wavelength, are usually fixed and cannot be easily varied, the roughness of the machined surface becomes more the outcome of the laser setup used rather than being a deliberate choice made by the operator. As an alternative approach for the control of surface roughness, advanced scanning algorithms [18] may be used. As the arrangement of individual craters on the surface is not influenced by the laser parameters mentioned earlier, multiple different scanning algorithms can be realized with the same laser setup. Therefore, the control of the surface roughness can be regarded as an intentional decision rather than a predetermined outcome of the laser setup used. It is evident that the latter method of controlling surface roughness is significantly more adaptable and versatile, making it highly significant. However, due to the lack of research in the field, there is still little knowledge of how different scanning algorithms influence the roughness of the surface. Additionally, as of today, no study has investigated the numerical evolution of surface roughness when using different scanning techniques.

In our study, we present an in-depth investigation of the scanning algorithms and how they influence the final surface roughness. We numerically investigate the evolution of the surface roughness when it is scanned with a femtosecond laser beam and compare the numerical results to the experimentally acquired values. The main aim of this study is to determine the optimal scanning technique for minimal surface roughness achievement after laser micromachining with femtosecond pulses. In the course of the study, a numerical model capable of predicting surface roughness after laser micromachining was created and verified experimentally.

## 2. Materials and Methods

### 2.1. Experimental Setup and Materials Used

The investigation of scanning techniques was conducted using two different laser sources: “Carbide” and “Pharos” (manufactured by Light Conversion Ltd., Vilnius, Lithuania), generating 200–300 fs duration pulses. Three different wavelengths, the fundamental (1030 nm), third harmonic (343 nm) and fourth harmonic (257 nm), were used to see the versatility of the determined scanning techniques. These were the only wavelengths that could have been generated using the available lasers. Different laser wavelengths enabled us to establish the relationship between the surface roughness and the used laser wavelength. Moreover, it allowed us to see if the scanning algorithms under investigation maintain when different laser wavelengths are utilized. The laser sources used in this study are depicted in Table 1. The carbide femtosecond laser shining 1030 nm (1H) wavelength radiation will be referred to as laser source number 1, while the carbide femtosecond laser with λ = 343 nm (3H) will be referred to as laser source number 2, and the Pharos femtosecond laser with λ = 257 nm (4H) will be referred to as laser source number 3. As can be seen from the table, different repetition rates and different pulse energies for different laser sources were used. The selected repetition rates for laser sources 2 and 3 correspond to the values at which the highest conversion efficiencies with the available harmonic modulus could be reached. For laser source 1, the repetition rate was chosen arbitrarily. As for pulse energies, the maximum achievable pulse energies with the available laser sources were chosen (Ep (1H) = 173 µJ; Ep (3H) = 10.8 µJ; Ep (4H) = 8.72 µJ) to generate reasonable-sized craters. For the same reason, the minimum available pulse durations were used (*t* (1H) = 300 fs; *t* (3H) = 300 fs; *t* (4H) = 220 fs). The properties of single-shot modifications (diameter and depth of craters) produced using different laser sources are also displayed in Table 1. All the experiments were performed on 1 mm thick soda-lime glass samples (typical microscope slides), with a primary surface roughness of Ra < 10 nm.

The transmission spectrum of the soda-lime glass sample was measured with a spectrophotometer (Shimadzu Corporation UVProbe) and is shown in Figure 1. From the graph, it is apparent that laser sources 1 and 2 operate in a transparency window, meaning that laser energy will be absorbed mostly non-linearly, while the laser light from source 3 will be absorbed in a linear manner.

The extensive study required a huge amount of data to be gathered and processed for the accurate determination of optimal scanning techniques. To save time and resources, a numerical model was developed that was capable of running surface ablation simulations and evaluating the arithmetical mean height (Ra) of the surface in the *x* and *y* directions. More about the construction of and working of the numerical model is presented in the section Theory and Calculation. To determine the accuracy of the model, a series of experiments were conducted using the presented laser sources.

The experimental setup consisted of one of the laser sources from Table 1, guiding optics (dielectric mirrors with high reflectivity for the used wavelength), a galvo-scanner (“SCANLAB exelliSCAN14”, “SCANLAB intelliSCAN14”, “SCANLAB intelliSCAN10_se_” for 1, 3, and 4 harmonics, accordingly) and a focusing lens (f-theta telecentric lens with focal lengths of 100 mm, 100 mm, and 160 mm for 1, 3, and 4 harmonics, accordingly). The sample was positioned using a motorized x/z axis system. The generalized optical scheme used in our study is presented in Figure 2. Each experiment was carried out by ablating 1 × 1 mm surface areas. The surface was scanned using different scanning algorithms. During the course of the study, three different scanning algorithms were investigated:Overlap of modifications in the *x* and *y* directions;The uncertainty of line positions;The shift of every second line.

All experiments were performed by positioning the top surface of the sample in the focal plane of the lens. This position in the z-direction was maintained during all ablation experiments. The visual surface inspection was realized using the Olympus BX51 optical microscope. For all surface roughness evaluations, machined areas were measured at 3 different locations using the Sensofar PLµ 2300 profilometer, and the average was taken as the result. No strict relocation method was used since the primary roughness of the surface was even over the whole area (<10 nm Ra). A single measuring window of 85 × 85 µm in size contained 256 profiles in the *x* and *y* directions each. The measured surfaces were plane-fitted to remove any tilt induced by possible sample positioning at an angle. For each profile, the Ra parameter [19] was calculated. To evaluate surface area roughness in one of the directions, an average of 256 profiles of a particular direction was taken. Although there are many different parameters for surface roughness evaluation, Ra is the most widely used in the literature. In pursuit of the ability to directly compare our results with others, the slightly modified version (average over the surface) of the parameter Ra was chosen as a base for the evaluations of surface roughness in the scope of this study. The areal Ra parameter (average of many profiles) considers the whole surface and not a single profile, thus providing more accurate results. Furthermore, utilizing this parameter allows for the assessment of surface roughness in various directions, which cannot be achieved by employing a single Sa parameter. For a better understanding of how this modified version of the parameter Ra is evaluated, see Figure 3. As can be seen in the B part of the figure, the surface roughness distribution in one of the directions is measured. It shows that due to induced structure on the surface (Figure 3A part), the surface roughness can vary quite significantly from profile to profile. To evaluate area surface roughness in the measured direction, the average roughness value of the distribution is taken as the result. The error of surface roughness is then the standard deviation of the roughness from its average value. Equations governing calculations of a roughness parameter for a single profile and the whole area are listed below (see Equations (Equation 1) and (Equation 2)).
(1)$$Ra_{x(j),y(j)}=\frac{1}{n}\sum_{i=1}^{n}\left|x_{i},y_{i}\right|$$
(2)$$Ra_{X,Y}=\frac{1}{N}\sum_{j=1}^{N}Ra_{x(j),y(j)}$$

Here, $x_{i}$ is the height deviation from an average value of a profile at the point *i*, $Ra_{x(j)}$ is the surface roughness of a *j* profile, and $Ra_{X}$ is the surface roughness of an area in a given direction. For the estimation of measurement noise, the subtraction technique was used [20]. From a set of 100 measurements, a noise value of Rq = 4.6 nm was estimated.

### 2.2. Theory and Calculation

A numerical model that analyses the evolution of surface topography during the laser ablation process was developed. The working principle of the model relies on the generated ablation depth dependence on the impinging fluence–material response function. This dependence was retrieved from the experimentally measured data. To retrieve the material response function, the fluence profile of the laser beam at the working distance (in our case, the focus position) and the geometry of the crater produced with such a laser beam had to be measured. To retrieve the laser beam fluence distribution at the working distance of the focusing lens, the imaging of the beam by a 4F system with a magnification of 6.6 times onto an Ophir SP503U camera was used. By simultaneously measuring laser pulse energy, we could normalize the camera fluence distribution measured by CCD so that the integral of the fluence profile would be equal to the measured pulse energy. The beam fluence profile measured and rescaled to its actual size is shown in Figure 4A. Using the measured laser beam, a series of single-shot craters were produced on the sample. The accuracy of the numerical model highly depends on how accurately the geometry of a single-shot modification can be measured. For this reason, the depth profiles of craters were measured using the Olympus LEXT OLS5000 laser scanning microscope, an instrument that had slightly better resolution than the Sensofar PLµ 2300 optical profilometer. However, since our access to the Olympus LEXT OLS5000 laser scanning microscope was very limited, it was used only for this single task; all other surface roughness measurements were performed using the Sensofar PLµ 2300 optical profilometer, as explained earlier.

The average of the depth profiles is shown in Figure 4A. As can be seen in the figure, the crater has rims that are up to 25 nm in height, steep edges, and a relatively flat bottom part. Once the laser beam profile and the depth profile of the crater were measured, a material response function was constructed (see Figure 4B). This dependence represents how much material is removed at a precise fluence value. After obtaining the material response function, numerical simulations of surface ablation could be conducted. As a starting point, a flat surface was chosen, as the measured soda-lime sample had a surface roughness of Ra < 10 nm, which should not influence the results. Knowing the beam fluence distribution of the laser and the material response function, the model was able to predict the change in the surface topography. The model considered the fluence change for the current pulse due to the slope of the surface induced by the previous pulses. When surface area increases, the fluence is reduced. Additionally, the model included the Fresnel reflection losses due to the laser radiation impinging the material not perpendicular to the surface. In the numerical simulations, the beam was translated over the surface by positioning the measured beam fluence distribution at different (manually selected) locations to modify the material. The path or precise locations of the pulses can be programmed beforehand. In such a way, different scanning algorithms were realized. The basic equations governing these steps can be written in the form:(3)$$H\left(x,y,N\right)=H\left(x,y,N-1\right)-h(x,y,F_{imp.}\left(x-x_{N},y-y_{N}\right))$$
(4)$$F_{imp.}=F\cdot L_{Fresnel}\cdot L_{proj.}$$
(5)$$L_{proj.}=\frac{1}{\sqrt{1+H_{x}{\left(x,y,N-1\right)}^{2}+H_{y}{\left(x,y,N-1\right)}^{2}}}$$
(6)$$L_{Fresnel}=1-R_{s}\left(H_{x},H_{y}\right)\cdot I_{s}-R_{p}\left(H_{x},H_{y}\right)\cdot I_{p}$$

Here, H(x,y,N) is the height value at the *x* and *y* coordinates after *N* number of pulses, $H_{x}$ and $H_{y}$ are partial derivatives in the *x* and *y* directions, and *h* is a function that returns how much of the material is removed at a certain fluence value with accordance to the material response function (see Figure 4B). $x_{N}$ and $y_{N}$ are the Nth pulse position on the surface, $F_{imp.}$ is impinging on the surface beam fluence, and *F* is the experimentally measured fluence. $R_{s}$ and $R_{p}$ are calculated Fresnel reflections [21] for impinging on an s or p polarization beam, which mostly depend on the surface gradient, while $I_{s}$ and $I_{p}$ define which polarisation beam is used for laser micromachining and have to follow the rule of $I_{s}$ + $I_{p}$ = 1, the ratio. Therefore, $I_{s}$/$I_{p}$ defines the ellipticity of the polarisation. Equation (3) is used for surface evolution calculations, while Equation (4) is used for calculating the reduction of laser fluence due to Fresnel reflections and the increase in surface area from previous shots. The principle scheme depicting the working of the numerical model is shown in Figure 5.

The incident beam fluence is shifted to a certain location on the surface (1). The reduction of the fluence due to Fresnel reflections (1.1) and the increase in surface area due to the craters from previous shots (1.2) are taken into account. From projected fluence and material response functions (2), the amount of ablated material is calculated (3). Finally, the amount of ablated material is subtracted from the initial surface (4). The incident laser beam is shifted to a different location (5), and the process repeats. As the positions of the beams can be chosen arbitrarily, the uncertainties of laser beam positioning were accounted for to imitate various vibrations and inaccuracies that were observed in the experimental setup. Moreover, when needed, due to a lack of synchronization between the laser and scanner systems, the uncertainty of the position of the first shot in a line was considered. After every simulation, the corresponding experiment was conducted, and surface roughness was calculated.

## 3. Results and Discussion

The numerical and experimental investigation was conducted by scanning the surface of the sample a single time using different scanning algorithms. It was observed that the surface roughness after a single pass is greatly determined by the overlap of modifications. By the term modifications, in this work, we refer to a damaged area on the surface of a sample (a.k.a. an ablated crater). It is important to notice that we are investigating the overlap of the damaged areas (which depend on incident energy) and not the overlap of the geometrical beams. Although the geometrical overlap is more commonly addressed in the literature [22,23], we propose that it is a parameter providing little information about the resulting surface morphology, and the overlap of modifications should be used instead. It is known that surface morphology after machining is a product of many overlapped modifications. Knowing only the spatial width of the beam (i.e., beam overlap) does not allow us to predict the geometry of the modification (more precisely, its width and edge steepness), as it depends on many different factors, such as beam shape, pulse duration, material response, etc. That being said, spatial beam overlap as a parameter for surface evaluation should be avoided, and overlap of modifications should be used instead. The overlap of modifications (Ω) was calculated using Equation (Equation 7) [23].
(7)$$\Omega =(1-\left(\frac{dx}{d_{mod.}}\right))*100$$

Here, dx represents the distance between modifications, and $d_{\mathrm{mod}.}$ represents the diameter of a single-shot modification. The overlap of modifications in the *x* and *y* directions were kept the same to realize even ablation. In Figure 6, we can see how surface roughness depends on the overlap of the modifications when the surface is scanned once using laser source number 3 (central wavelength—257 nm). It can be observed that the results of numerical simulations (black curve) and experimentally acquired values (red curve) are in good agreement. The lowest surface roughness is achieved when the overlap of modifications is in the range of 10–20%. If such conditions are satisfied, the surface roughness of Ra∼ 20 nm can be achieved. From the graph, it appears that even lower surface roughness values can be achieved if beam distance is increased by more than 30 µm (−100% overlap). However, it is just a measurement artifact. When the beam distance is drastically increased, modifications are placed further and further apart to the point where the distance between single-shot modifications exceeds the dimensions of the measuring window. In that case, the primary surface roughness of a sample is measured (Ra < 10 µm), creating an illusion of excellent results. On the other end of the curve, we can see an increase in surface roughness with an increase in the overlap of the modifications. When the overlap of modifications was increased by more than 80%, the soda-lime glass samples started to shatter due to induced stress in the surface. For the stated reasons, such a regime is not applicable and will no longer be discussed in the scope of this study. After a more detailed examination of the results of numerical calculations, the periodic variations in surface roughness can be noticed (see Figure 6, black curve, in the region where the beam distance is 3–15 µm). As discussed above, the minimum roughness is achieved when the distance between consecutive modifications is 13 µm (or 13.3% overlap); however, other local minimums in the curve can be seen at beam distances of 6.3 µm and 3.2 µm. These results suggest that an optimal arrangement of modifications is achieved not only at a beam distance of 13 µm but also at twice smaller distances of 6.3 µm and four times smaller distances of 3.2 µm. It is likely that this pattern continues further. However, with each iteration (when the beam distance is halved), the roughness of the surface increases. In contrast, the experiments did not show the same variation in surface roughness as was observed in the case of numerical calculations. From the steepness of the dips in the curve, it can be assumed that highly precise pulse positioning is necessary for optimal arrangements to be established. However, in reality, all sorts of vibrations and inaccuracies exist that likely restrict the formation of such patterns; therefore, they were not observed experimentally. To further address this observation, a pulse positioning system of high accuracy is necessary. However, we do not have access to a system with such accuracy. As a result, it remains unclear whether it is possible to replicate the observed variations in surface roughness experimentally.

Additional experiments were carried out with different laser systems to test if the determined relation holds. The results are shown in Figure 7. The dependence of surface roughness on the overlap of the modifications when different wavelengths are used is shown. Looking at the figure, few conclusions can be drawn.

It appears that lower surface roughness can be achieved when a shorter wavelength laser light is used. The absorption coefficient is highest for the shortest wavelength radiation; thus, the light is absorbed in the smallest depth, resulting in more gentle ablation (see Figure 8B) and lower surface roughness. On the other hand, for the same reason, the ablated depth is smallest when using the shortest wavelength radiation (see Table 1). This shows that it is possible to achieve lower surface roughness using shorter wavelengths but at a trade-off of lower ablated depths. Additionally, it can be seen that the minimum achievable surface roughness can be reached when the overlap of the modifications falls between 10 and 25% for all the different laser sources regardless of the wavelength, used pulse energy, and crater diameter. To better understand why the best surface quality is achieved when the modifications are just barely overlapped, it is worth looking into the geometry of a single-shot modification (see Figure 8B). One of the most important parts of the modification geometry that greatly determines the resulting surface roughness is the steepness of the edges. From the graph, we see that the edges of produced modifications are steep. It is believed that steep edges are the result of pulse duration, specifically of femtosecond pulses [24,25]. When modifications are overlapped, a fraction of previously generated modification is ablated by the new coming one. To realize the smoothest surface, the modifications have to be overlapped so much that only the steep edges would be ablated, and relatively flat bottoms would remain undamaged. It appears that such conditions are satisfied when modifications produced by femtosecond pulses are overlapped by 10–25%.

More parameters describing the surface roughness properties of the surfaces at optimal overlap values can be found in Table 2. The roughness parameters provided demonstrate a close correlation between our proposed areal Ra parameter and the conventional Sa (surface arithmetical mean height) parameter. Moreover, it can be seen that if the surface has different roughness values measured in different directions (such a case happens when the laser system is not synchronized), the Sa parameter is determined by the more rough direction of the surface. Hence, in a case when only the Sa parameter is measured, only half of the information about the roughness of the surface is retrieved. Therefore, measuring $Ra_{X}$ and $Ra_{Y}$ separately provides more information about the roughness of the surface than a stand-alone Sa parameter. Other parameters, such as skewness (Ssk) and kurtosis (Sku), indicate that all three surfaces have a relatively even distribution of peaks and valleys over the area. Moving on, several advantages of using UV light in the context of surface roughness can be noted. When looking at Figure 8C, we can notice that the material response function for the 4H is shifted to the lower fluence values when compared to the 3rd and 1st harmonics. This is believed to be the result of linear absorption [26].

This enables us to produce modifications of the same depth when compared to 3H and 1H but at 3 times lower fluence values. This means that more energy is used for ablating the material and less for heating it; thus, the heat-affected zone is minimized, and modifications of better quality can be produced. This also shows that when using a 4H harmonic for soda-lime glass laser machining, the ablation threshold is achieved at lower fluences when compared with 1H and 3H, meaning there is no need for high-power laser sources that can be dangerous and challenging to work with. This observation agrees with other works found in the literature [27].

Another important insight that was made during the study is that if the beam guiding system is not synchronized with the laser, the modified surface roughness in the *x* and *y* directions can differ quite significantly. In a non-synchronized system, the laser and scanner are not synchronized. The laser generates pulses at a fixed (manually selected) repetition rate. To create desired scanning paths on the surface of the sample, the triggering of the pulse picker of the laser is controlled via an external signal (using Scanlab RTC5 control board). The pulse picker trigger signal does not account for the time position of the nearest generated pulse within the laser; therefore, when the pulse picker is triggered (e.g., at the starting position of the line), the time delay after which the first pulse will arrive is random. This introduces uncertainty to the position of the first pulse in the line. The magnitude of the uncertainty is determined by the scanning speed and the repetition rate of the laser. The shift (Δ) can be evaluated using Equation (Equation 8).
(8)$$\Delta =\frac{v}{f}=dx$$

Here, *v* represents the scanning speed, *f* represents the laser repetition rate, and dx represents pulse distance. The laser ablated surface for two cases when the system is synchronized and not synchronized are depicted in Figure 9. We see that when the beam guidance system is synchronized, all the modifications are nicely arranged in a rectangular manner. On the other hand, when the system is not synchronized, all lines are shifted. While this does not affect surface roughness in the scanning direction (*x* direction), the surface roughness in the *y* direction changes. This can be observed from the dependence of the surface roughness on the overlap of modifications in the *x* and *y* directions. The surface roughness in the *y* direction is usually higher. The measured difference was as high as 37% (when the overlap was 5.4%). Since many different material parameters depend on the surface roughness, the difference in surface roughness in the *x* and *y* directions can be exploited. One of the possible applications would be the control of the movement direction of fluids and gases [28,29].

Last but not least, it was shown that surface roughness can be further improved by shifting every second line by a certain distance in the scanning direction. In Figure 10, a dependence of surface roughness on the shift of every 2nd line is depicted. The shift of every 2nd line was normalized to the pulse distance and plotted on the *x*-axis. It can be observed that the lowest surface roughness is achieved when every second line is shifted by half of the pulse distance. By realizing such conditions, surface roughness can be further reduced by an additional 20%. When every second line is shifted by half a distance between single pulses, a checker pattern is realized. A small portion of unablated material that is usually left in a cross-section of 4 pulses (see Figure 6 topographies) is now being ablated while still maintaining the overlap of modifications close to optimal. If such conditions are met, further reduction of surface roughness is possible.

After a thorough investigation, it was demonstrated that surface roughness can be tailored or minimized by choosing an appropriate scanning algorithm. A detailed explanation has been introduced for the first time, establishing the underlying reasons for the attainment of minimum surface roughness when the overlap of craters is within the range of 10–25%. In addition, the relationship between surface roughness and used laser wavelength was demonstrated and reasoned. Next, a novel technique for attaining diverse levels of roughness in distinct orientations was introduced and discussed. A new perspective for evaluating the areal surface roughness was introduced, offering areal surface roughness measurements based on its orientation. Furthermore, a novel scanning algorithm was introduced that involves shifting every second line and has the capability to achieve an even greater reduction in surface roughness.

During the course of the study, a numerical model based on experimentally measured data was constructed. It was shown that the model is capable of running surface ablation simulations and accurately predicting final surface roughness. To the best of our knowledge, this is the first numerical model of such a kind that accounts for the Fresnel reflections and fluence reduction due to the increase in surface area and allows for such an accurate estimation of final surface roughness under the provided conditions.

However, the study was limited by investigating only single-scanned surfaces. The achieved depths after a single scan were very low (usually <1 µm). In practice, to retrieve any shape using laser ablation, the surface must be scanned multiple times to remove a reasonable amount of material. This case, when the surface is scanned multiple times, is usually of more interest. With the aforementioned rationale in mind, our future research is focused on studying how to control surface roughness when scanning the surface multiple times. We are curious to further explore the ability to control the roughness of the surface through the use of different scanning algorithms. The constructed numerical model should be of huge assistance in future works when multi-scan ablated surfaces will be investigated. Another limitation of this study was that only surfaces with very low (Ra < 10 nm) primary surface roughness were investigated. It is yet unknown how the roughness of the surface would be affected if the determined scanning algorithms would be applied to ablate the surface of high primary surface roughness. Since only a minority of the materials have a very low primary surface roughness (such as the samples used in this study), the ablation of rough surfaces and the control of the surface roughness of such materials is of huge interest. Therefore, due to the lack of knowledge in the field, investigating this direction appears to be another promising avenue for future research.

## 4. Conclusions

After a thorough investigation, the following conclusions were drawn:It was determined that when the surface is scanned once, final surface roughness is greatly determined by the overlap of the modifications and not greatly determined by the geometrical beam overlap. It was seen that the smoothest surface is realized when the overlap of modifications is in the range of 10–25% (Ra− 20 nm). This tendency was maintained even when different laser sources were used. Under such conditions, only steep edges were ablated, while relatively flat bottoms were left undamaged, resulting in a smooth surface.It was observed that in the case of a non-synchronized system (due to a lack of synchronization between the laser and the scanner), the final surface roughness in the *x* and *y* directions is different. The measured difference was as high as 37%. The variation in surface roughness according to surface orientation is attributed to the formation of distinct structures in varying directions on the surface.It was shown that the further reduction of surface roughness is possible if the scanning technique of a checker pattern is used. When every 2nd line is shifted by half a step between adjacent modifications, the un-ablated material that is usually left in a cross-section of 4 modifications is now being ablated, resulting in a further reduction in final surface roughness.

Based on the preceding conclusions, it is evident that surface roughness can be regulated (minimized) by selecting appropriate scanning algorithms. Moreover, the surface roughness dependencies on the scanning algorithms determined in the study proved to be independent of the laser parameters, therefore enabling the modification of surface roughness exclusively through the choice of scanning algorithms regardless of the configuration of the laser setup.

## Figures and Tables

**Figure 1 materials-16-02788-f001:**
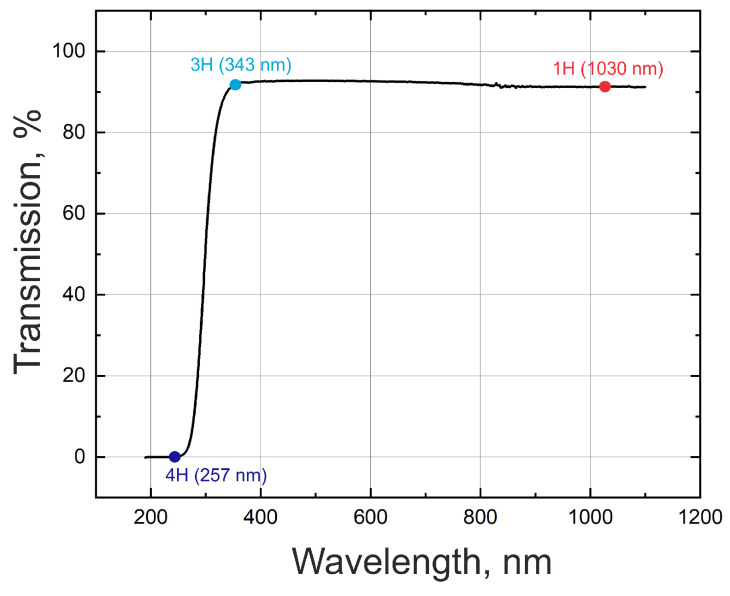
Transmission spectrum of a 1 mm thick soda-lime glass sample.

**Figure 2 materials-16-02788-f002:**
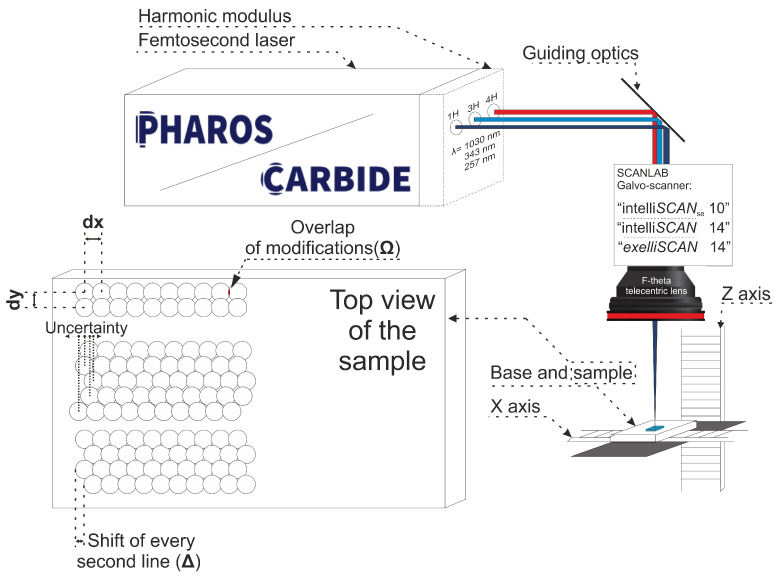
Principle scheme of the laser system used in the study. Femtosecond lasers “Carbide“ or “Pharos“ was used as the laser source. The beam was guided via dielectric mirrors of high reflectivity for a specific wavelength. The beam was scanned on the surface of the sample using the SCANLAB galvo-scanner and f-theta lens combination. The sample was placed on two axes allowing movement in *x* and *y* directions. In the left bottom part of the picture, the top view of the sample and an explanation of different notations used in the study are presented. dx—distance between single modifications in *x* direction, dy—distance in *y* direction, Ω—overlap of modifications, Δ—shift of every second line.

**Figure 3 materials-16-02788-f003:**
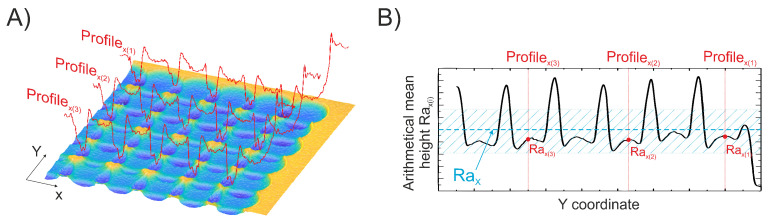
Scheme explaining how surface roughness is evaluated. The topography of a sample after single scan machining, when the overlap of the modifications is low, is shown in part (**A**). Red curves represent profiles at discrete locations. Surface roughness dependence on *Y* coordinates is depicted in part (**B**). $Ra_{x(j)}$ is a roughness parameter of a *j* profile. The roughness of an area is then the average ($Ra_{X}$). Blue dashed line shows the position of $Ra_{X}$ on the y-axis, where tilted blue lines represent the range of one standard deviation.

**Figure 4 materials-16-02788-f004:**
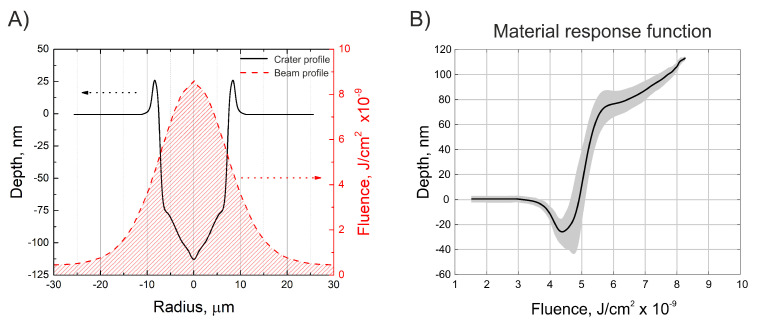
Fluence profile of the laser beam at its focus position, measured and rescaled to its actual size, and the measured geometry of the crater produced with a such laser beam (**A**). Constructed material response function, which shows how much material is removed at precise fluence value (**B**). The black line represents the average value, whereas the grey area shows the boundaries of standard deviation. The laser source used was number 3 (see Table 1).

**Figure 5 materials-16-02788-f005:**
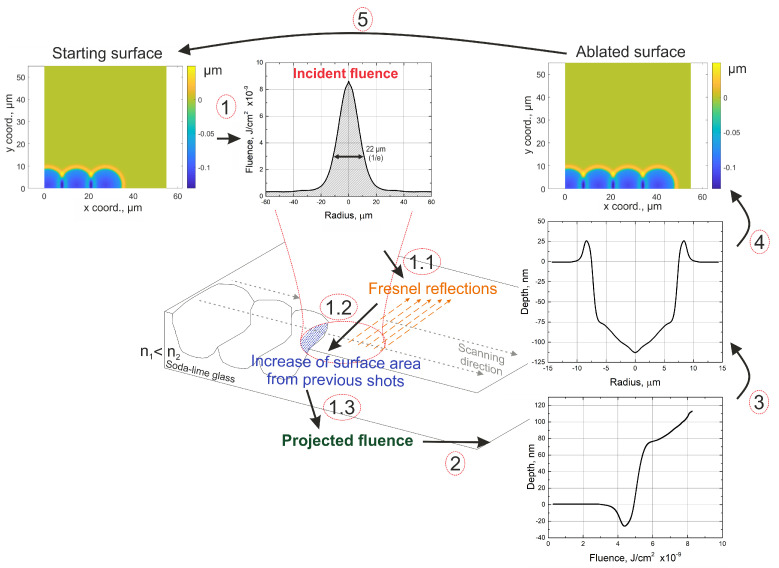
Principle scheme showing the working of the numerical model. The incident fluence is projected on the surface (1). Fluence reduction due to Fresnel reflections (1.1) and increase in surface area from previous shots (1.2) is taken into account. The projected fluence is then calculated (1.3). According to the material response function and projected fluence (2), the ablated volume is calculated (3). Finally, the ablated volume is subtracted from the surface at the current beam location (4). The position of the beam is shifted, and calculations are repeated (5).

**Figure 6 materials-16-02788-f006:**
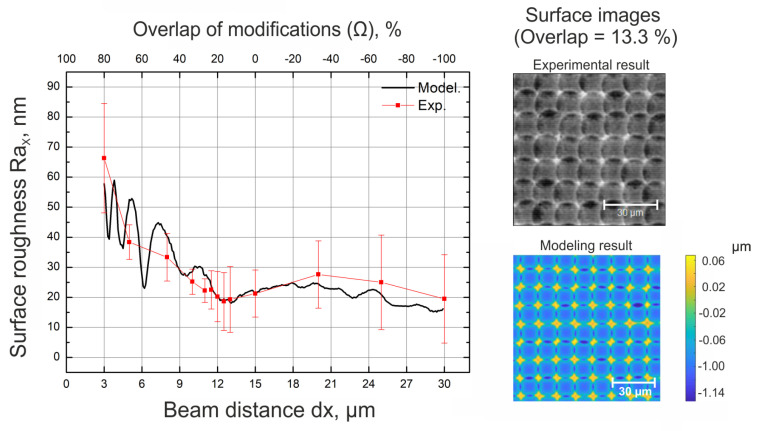
Surface roughness (in the scanning direction) dependence on the overlap of modifications (on the **left**). Surface images after single-pass machining acquired experimentally and via numerical calculations when the overlap of modifications is 13.3 % (on the **right**). The laser source used was laser source number 3 (4H).

**Figure 7 materials-16-02788-f007:**
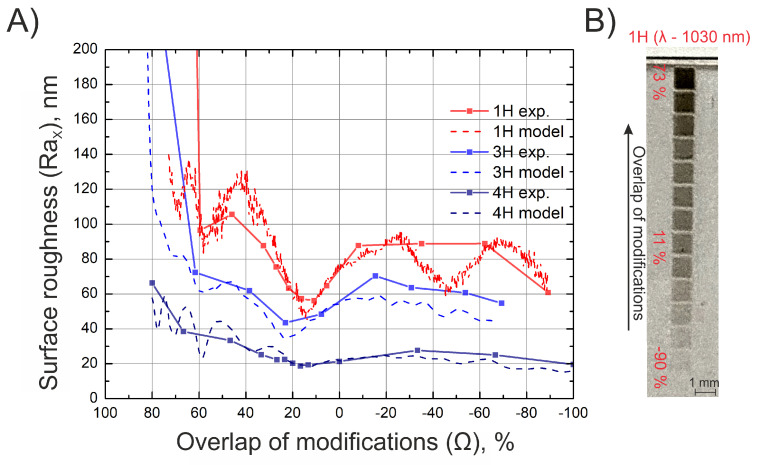
Surface roughness dependence on the overlap of modifications when different laser wavelengths are used (**A**). Surface roughness measured in scanning (x) direction. Solid lines represent experimental results, and dotted lines represent modeling results. In (**B**), a macro picture of machined surfaces (using laser source 1) is shown.

**Figure 8 materials-16-02788-f008:**
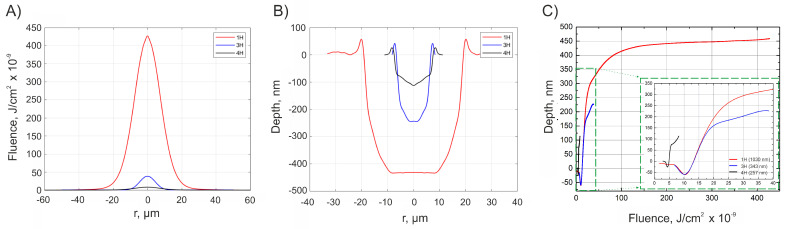
Beam profiles using different laser wavelengths (**A**), profiles of single-shot modifications using different laser wavelengths (**B**), and material response functions when different wavelengths are used (**C**). The subfigure in part (**C**) provides a closer look of material response functions at lower fluence and depths values. Red curve—1H, blue curve—3H, black curve—4H.

**Figure 9 materials-16-02788-f009:**
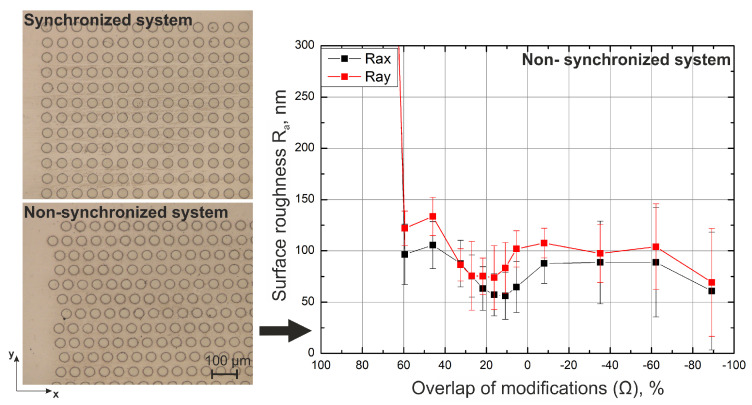
Picture of the surface after a single-pass ablation when the system is synchronized (on the **top left** picture). Picture of the surface after a single-pass ablation when the system is not synchronized (on the **bottom left** picture). Surface roughness dependence on the overlap of the modifications when the system is not synchronized (on the **right**). Used laser source—number 1.

**Figure 10 materials-16-02788-f010:**
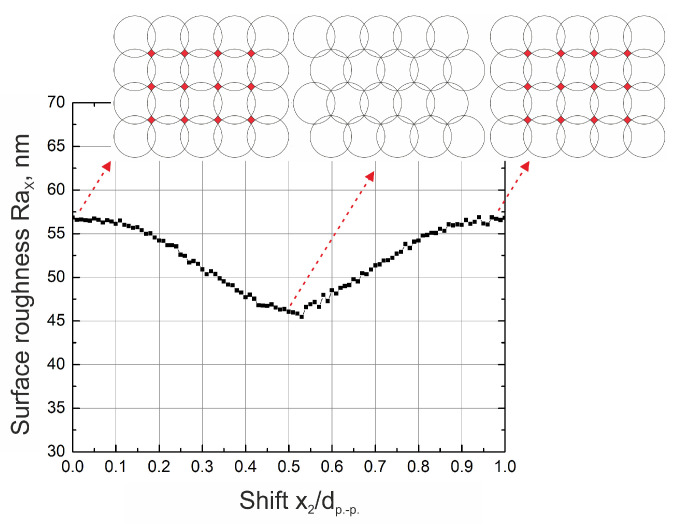
Surface roughness dependence on the shift of every 2nd line when pulse distance of 32 µm is maintained. On the *x*-axis, the shift between consecutive lines $x_{2}$ divided by the pulse distance is plotted. Used laser system—number 1. Surface roughness in *x* direction. Data of numerical simulations.

**Table 1 materials-16-02788-t001:** Parameters of different laser sources used in the study and the properties of single-shot modifications.

	Laser	Wavelength	Pulse Duration (FWHM)	Pulse Energy	Repetition Rate	Diameter of Laser Beam at Focal Plane (at 1/e high	Diameter of Single-Shot Modification	Depth of a Single-Shot Modification
Source 1	Yb:KGW laser source “Carbide”	1030 nm (1H)	300 fs	173 µJ	200 kHz	21.4 µm	38 µm	432 nm
Source 2	Yb:KGW laser source “Carbide”	343 nm (3H)	300 fs	10.8 µJ	602 kHz	12.6 µm	13 µm	226 nm
Source 3	Yb:KGW laser source “Pharos”	257 nm (4H)	220 fs	8.72 µJ	50 kHz	22 µm	15 µm	113 nm

**Table 2 materials-16-02788-t002:** Different surface roughness parameters of experimentally produced surfaces using different laser sources at optimal overlap values. $Ra_{X}$ and $Ra_{Y}$—the arithmetic average height of a line averaged over the area in *x* and *y* directions, accordingly, Sa—surface arithmetic average height, Sq—RMS roughness of the surface, Ssk—surface skewness, Sku—surface kurtosis [19].

	${\mathrm{Ra}}_{X}$, nm	${\mathrm{Ra}}_{Y}$, nm	Sa, nm	Sq, nm	Ssk, nm	Sku, nm
1H, Ω = 10.8%, non-synchronized	55.3	80.42	82.2	107.5	0.0887	0.572
3H, Ω = 23%, non-synchronized	42.61	62.07	62.17	84	−0.085	1.316
4H, Ω = 16.6%, synchronized	19.33	19.87	21.96	30.19	0.515	1.661

## Data Availability

Any further details relevant to this study may be obtained from the authors upon reasonable request.

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
