# Peer review of "Scanning Algorithm Optimization for Achieving Low-Roughness Surfaces Using Ultrashort Laser Pulses: A Comparative Study"

_materials, 2023, doi:10.3390/ma16072788_

Round 1

Reviewer 1 Report

I can recommend the publication of this manuscript after a minor revision.

1. Insert references for all mathematical formulas.

2. Line 75: Re-formulate the sentence: “...predicting resulting surface roughness”. Surface roughness is not expressed only by the Ra parameter!

3. Line 93: Re-formulate the sentence: “For surface roughness evaluation...”. Surface roughness is not expressed only by the Ra parameter! Also, in other sentences from the manuscript.

4. Replace in fig. 3, “Surface roughness” with the name of the calculated parameter (Arithmetical mean height Ra).

5. Line 127: specify the calculation method applied for “Fluence” (mathematical formulas, if possible).

6. Explain with more details sentences from lines: 135-137, 175-177, 236-238.

7. Line 150: minor mistake “Pinciple”.

8. A low-roughness surface is not expressed only by Ra parameter. If possible, could you specify some microtexture parameters for the samples such as: Root mean square height Sq [nm]; Skewness Ssk [-]; Kurtosis Sku [-]?

9. Specify the limits of this study. State in more detail the respective advantages and disadvantages.

10. Insert Data Availability Statement.

11. Re-formulate the title of the manuscript.

This manuscript can be published after the mentioned revisions.

Reviewer 2 Report

The authors propose a new method for predicting resulting surface roughness through material response function obtained by experiment. Based on the numerical simulations, the influence of overlap of the modifications, wavelength of fs laser and scanning algorithms on roughness were investigated. However, the authors should consider the following comments/questions:

(1) The error bands should be required in material response function in Fig. 4 to rule out contingency.

(2) The depth units in Line 123 do not match with the units in Fig 4, please double check.

(3) A detailed explanation is required for the reason of periodic variations in roughness when the beam distance is in the range of 3-9 μm, as shown by the black line in Fig.6.

(4) The pulse duration and pulse energy of three fs-laser are different in the manuscript in Table 1. Besides, focal lengths of f-theta telecentric lens is also different in Line 82. Thus, the multiple variables need to be excluded to ensure the accuracy of the comparison of the results of roughness in Fig. 7. The different profiles of ablated craters and laser beams need to be displayed when three different laser sources were used.

Reviewer 3 Report

Dear Author(s), the manuscript ‘Numerical simulations and experimental analysis of scanning algorithms to achieve low-roughness surfaces using ultrashort laser pulses’, Manuscript ID: materials-2298299, have some weakness that must be revised suitably.

Please find below some, of the most significant comments:

1.      The ‘Abstract section’ must be improved so that, firstly, the ‘low surface roughness depends on its applications. For some cases, 5 µm is not ‘low’ roughness. Moreover, the Ra parameters are more on the profile roughness than the surface (Sa).

2.      Secondly in the ‘Abstract’ section, the abbreviations should be omitted. What is ‘fs pulses’ for the reader of the Abstract section only?

3.      As another issue, motivation in the Abstract is difficult to improve according to the general proposals. The Author(s) should introduce the issue and then, respectively, propose some solutions.

4.      From the ‘Introduction’ section, the motivation, presented in lines 45-52 does not derive from the critical review in the previous gaps of this section. In fact, a critical review was not provided and the proposals do not respond to the lack of the current state of knowledge. From that matter, the meaning of motivation can be lost.

5.      The values of the parameters and especially, their ranges, must be justified. In its current form looks like selected arbitrarily.

6.      Some details on surface (profile) roughness measurement are in lines 92-108 and Figure 3. However, it is not clear, how the relocation method validation was received. Moreover, the precision of the re-location is not improved. If it is, must be indicated more clearly.

7.      The reduction of measurement uncertainty is not the same as the precision of the relocation method. It must be appropriately presented and separated for each of the issues.

8.      According to the previous comment, there is no word on the received data validation. If uncertainty is included, it was not appropriately mentioned. Moreover, there is no word against measurement errors, noise, etc. Please try to refer to those issues like in:

(1)   https://doi.org/10.1088/2051-672X/3/3/035004

(2)   https://doi.org/10.24425/mms.2020.132772

(3)   https://doi.org/10.1016/j.cirp.2014.03.086

9.      Why for surface or profile inspections there were used two devices, Olympus BX51 optical microscope and Olympus LEXT OLS5000 laser scanning microscope? Increasing the number of measurement devices the number of measuring and data processing errors can also be enlarged. It is not justified or even motivated (limited).

10.  All of the equations used in the manuscript should be referenced to the primary sources that, correspondingly, are not newly proposed by the Author(s).

11.  In section no.3, including discussion, in fact, there is no critical discussion, especially presenting the lack of the methods (analysis) provided. Any weaknesses of the studies are not allowed by the Author(s).

12.  According to the above comment, there is no proposal for ‘Future prospects’ or ’The outlook’ for the studies. Not to mention the weak parts of the manuscript, future proposals would not exist.

13.  The ‘Conclusions’ section should be divided into separated and numbered gaps which would be helpful in the definition of the novelty, which should be highlighted more clearly.

14.  One general proposal and conclusion must be provided in the last section as well. The reader should be convinced about one main proposal of the study. In many cases, the reader feels lost and difficult to follow what the Author(s) are trying to convey.

15.  Concluding, from the manuscript, it is difficult to receive one, general idea that the reader is trying to learn. Please try to emphasize the main novelty of the paper. Usually, excluding many ideas, one is this crucial, pls highlight it.

From the above, the reviewed manuscript must be improved significantly before any further processing, if allowed by the Editor.

Round 2

Reviewer 3 Report

Dear Author(s), the manuscript titled ‘Numerical simulations and experimental analysis of scanning algorithms to achieve low-roughness surfaces using ultrashort laser pulses’, Manuscript ID: materials-2298299, has been improved according to the minimum requirements so, if allowed by the Editor, can be further processed by the Materials journal.